# Changes in HbA$_{1c}$ during the first six years after the diagnosis of Type 2 diabetes mellitus predict long-term microvascular outcomes

**Maarten P. Rozing** [ORCID]\*, **Anne Møller, Rune Aabenhus, Volkert Siersma, Katja Rasmussen, Rasmus Køster-Rasmussen**

The Research Unit for General Practice and Section of General Practice, Department of Public Health, University of Copenhagen, Copenhagen, Denmark

\* mroz@sund.ku.dk

**Data Availability Statement:** The data underlying the results presented in the study are available from the Research center for general practice,

## Abstract

To analyze the association between change in HbA$_{1c}$ during the first 6 years after diagnosis of Type 2 diabetes mellitus (Type 2 DM) and incident micro- and macrovascular morbidity and mortality during 13 years thereafter. This is an observational study of the participants in the intervention arm of the randomized controlled trial Diabetes Care in General Practice (DCGP) in Denmark. 494 newly diagnosed persons with Type 2 DM aged 40 years and over with three or more measurements of HbA$_{1c}$ during six years of intervention were included in the analyses. Based on a regression line, fitted through the HbA$_{1c}$-measurements from 1 to 6 years after diabetes diagnosis, glycaemic control was characterized by the one-year level of HbA$_{1c}$ after diagnosis, and the slope of the regression line. Outcomes were incident diabetes-related morbidity and mortality from 6 to 19 years after diabetes diagnosis. The association between change in HbA$_{1c}$ (the slope of the regression line) and clinical outcomes were assessed in adjusted Cox regression models. The median HbA$_{1c}$ level at year one was 60 (IQR: 52–71) mmol/mol or (7.65 (IQR: 6.91–8.62) %). Higher HbA$_{1c}$ levels one year after diagnosis were associated with a higher risk of later diabetes-related morbidity and mortality. An increase in HbA$_{1c}$ during the first 6 years after diabetes diagnosis was associated with later microvascular complications (HR per 1.1 mmol/mol or 0.1% point increase in HbA$_{1c}$ per year; 95% CI) = 1.14; 1.05–1.24). Change in HbA$_{1c}$ did not predict the aggregate outcome 'any diabetes-related endpoint, all-cause mortality, diabetes-related mortality, myocardial infarction, stroke, or peripheral vascular diseases. We conclude that suboptimal development of glycaemic control during the first 6 years after diabetes diagnosis was an independent risk factor for microvascular complications during the succeeding 13-year follow-up, but not for mortality or macrovascular complications.

## Background

In general, persons with Type 2 diabetes mellitus (Type 2 DM) have an increased cardiovascular morbidity and mortality [1, 2], and an elevated haemoglobin A$_{1c}$ (HbA$_{1c}$) increases the risk

Copenhagen university for researchers who meet the criteria for access to confidential data (Dagny Ros Nicolaisdottir, datamanager, Copenhagen University; email address: dagny.ros@sund.ku.dk). When requesting access to the data underlying our study, future researchers should file a request for the "DCGP data set".

**Funding:** Major funding for this study was received from The Danish Medical Research Council, The Danish Research Foundation for General Practice, The Health Insurance Foundation, The Danish Ministry of Health and The Pharmacy Foundation. Additionally, funding was received from the commercial source Novo Nordisk Farmaka Denmark Ltd. The funders had no role in study design, data collection and analysis, decision to publish, or preparation of the manuscript.

**Competing interests:** Funding was received from the commercial source Novo Nordisk Farmaka Denmark Ltd. This does not alter our adherence to PLOS ONE policies on sharing data and materials. There are no patents, products in development or marketed products to declare.

of both micro- and macrovascular complications in these persons [3–5]. Randomized controlled trials have shown that treatment with anti-diabetic medication lowers the risk of complications[6]. It is unknown if the lower risk associated with anti-diabetic treatment is fully attributable to the lowering of HbA$_{1c}$ levels [7], or that other pleiomorphic factors, such as the pharmaceutical properties of the anti-diabetic drugs, account for this effect.

The Diabetes Care in General Practice (DCGP) study was a cluster randomized controlled trial assessing the effect on mortality and morbidity of structured personal care compared with routine care in a population-based sample of persons newly diagnosed with clinical Type 2 DM in Denmark [8]. After six years, the structured personal care group had improved glycaemia, blood pressure, total cholesterol, and microalbuminuria. There were no differences in terms of morbidity and mortality between the two groups [8]. However, a 19-year follow-up of the DCGP study showed that participants in the intervention group had a lower incidence of myocardial infarction and 'any diabetes-related outcome', while mortality rate was similar between the two groups [9]. The intervention group displayed a large inter-individual variation in HbA$_{1c}$ [10]. On average, an initial drop in HbA$_{1c}$ was observed, followed by an increase during the remaining five years of the intervention period. Only age and HbA$_{1c}$ at time of diagnosis were predictive of the variation in HbA$_{1c}$ during follow-up [10].

Earlier studies have used a single HbA$_{1c}$ measurement [5] or an average of several HbA$_{1c}$ measurements [11] to predict later outcomes. In the present confirmatory study, the aim was to determine the relation between changes in HbA$_{1c}$ during 6 years of intervention in persons with newly diagnosed Type 2 DM and any diabetes-related endpoint, all-cause mortality, diabetes-related mortality, myocardial infarction, stroke, peripheral vascular disease and microvascular disease during the 13 years thereafter.

## Participants and methods

### Study design and population

This is a cohort study of the intervention group from the DCGP study, a cluster randomized controlled trial. Informed consent was obtained from all participants. The protocol was in accordance with the Helsinki declaration and was approved by the ethics committee of Copenhagen and Frederiksberg (V.100.869/87). All data were fully anonymized before they were accessed. In 1988, 474 general practitioners volunteered to take part in the study. These were randomized to structured personal care or routine care [8]. All doctors were to include all participants aged 40 years or over with newly diagnosed diabetes (Fig 1). The treatment guidelines in the intervention group included follow-up every three months and individualized goal setting for each participant supported by prompting of doctors, printed guidelines, feedback, and continuing medical education. The control group was not included in the present analysis, as yearly HbA$_{1c}$ measurements were performed only in the intervention group. Hence, this study is confined to the 536 participants who were alive and were examined at the end of the intervention 6 years after the diabetes diagnosis. Participants with less than three HbA$_{1c}$ measurements between 1 year after diagnosis and 6-year follow-up were excluded. The final study sample included 494 participants (Fig 1). As only 2.5% of the cohort started insulin treatment within 180 days after diagnosis, we consider that 97.5% had Type 2 DM [12].

### Definition of glycaemic control (determinants)

At the time of diabetes diagnosis and at the annual examinations, the fraction of HbA$_{1c}$ was determined by the same ion-exchange, high-performance liquid chromatography method with a relatively high reference interval (5.4–7.4% (36–57 mmol/mol)) at Odense University Hospital [8]. In general, participants experienced an improved glycemic control with a more

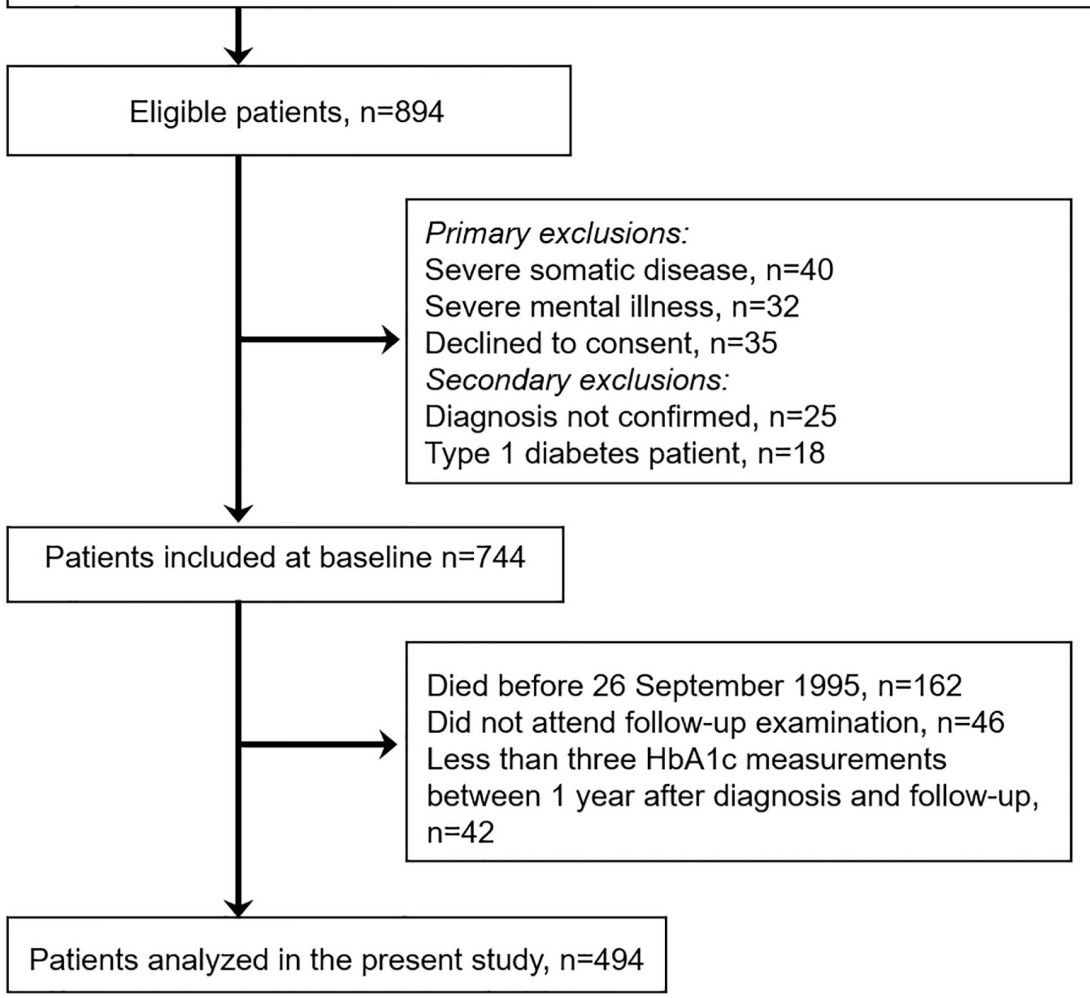

Fig 1. Flow chart depicting the number of participants in the Diabetes Care in General Practice (DCGP) study included in the current study.

or less pronounced remission of hyperglycaemia during the first year after diabetes diagnosis [10, 13]. To determine the individual development in glycaemic control, we fitted linear regression lines to the HbA1c measurements from one to (on average) six years after the diagnosis for each participant individually. Glycaemic control was then characterized for each

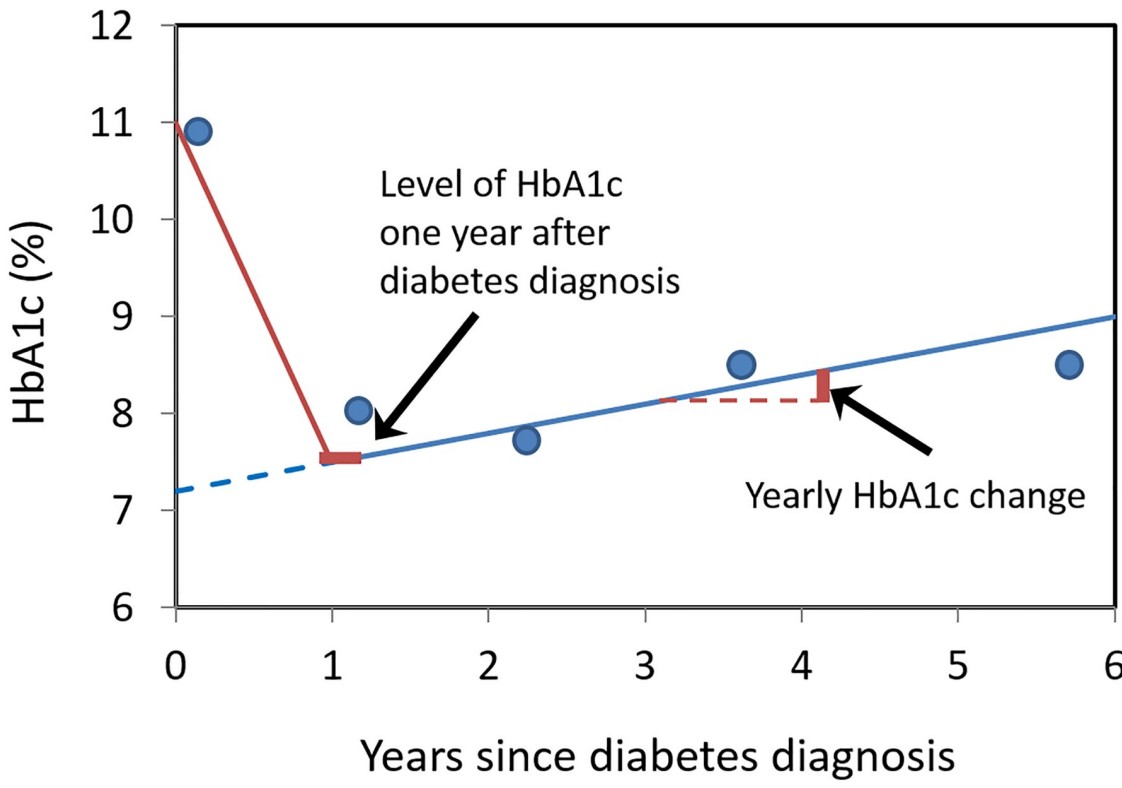

**Fig 2. Graphic depiction of the covariate LEVEL at year one of the 6-year intervention period after diagnosis and the SLOPE of the change in HbA$_{1c}$ from year 2 through 6 year of the intervention period.**

individual participant from this regression line, firstly, by the one-year intercept of the regression line ("one-year HbA1c level") and, secondly, as the slope of the regression line ("yearly HbA1c change") [10] (see Fig 2).

## Definition of outcomes

The outcomes were all-cause mortality, diabetes-related mortality (i.e. cardiovascular mortality, death due hypo- and hyperglycemia, and sudden death) myocardial infarction, stroke, peripheral vascular disease, and microvascular disease, any diabetes-related endpoint (i.e. presence of any of the aforementioned categories) (please see S1 Table for the exact definitions of the different outcomes). If a participant had an outcome at the time of diabetes diagnosis, the participant was excluded from the analyses of that specific outcome. Outcomes were assessed in the Danish registries until 31 December 2008. The vital and emigration status of all participants were ascertained through the Danish Civil Registration System [14]. Information on morbidity and causes of death were retrieved from the National Hospital Discharge Registry [15] and Danish Register of Causes of Death [8].

## Covariates

Covariates were selected based on measured confounders known to affect both mortality and HbA$_{1c}$. Information about covariates was collected at 6-year follow-up [10]. Age, sex, and cohabitation status (dichotomous: living alone yes/no) were registered. Body mass index (BMI, kg/m$^2$) was calculated based on height measured at the time of diagnosis and weight measured at the end of the intervention, on average 6 years later, by the general practitioner

with the scales available in the clinic. Wasting/pathological weight loss was defined as unintentional weight loss $\geq$ 1 kg per year (wasting yes/no) (16). Current smoking status was dichotomized (yes/no). Information about leisure time physical activity (LTPA) was assessed from the International Physical Activity Questionnaire and dichotomized as "sedentary LTPA yes/no" [16]. Hypertension was defined as systolic blood pressure $\geq$ 160, and/or diastolic pressure $\geq$ 90 mmHg, and/or antihypertensive and/or diuretic treatment. Laboratory tests included total cholesterol (mmol/l), fasting triglycerides (mmol/l), serum creatinine (μmol/l), and urinary albumin (mg/l). Type of treatment at the end of intervention was categorized as: diet only, oral antidiabetic, or insulin.

## Statistical methods

The effect of one-year HbA$_{1c}$ levels was expressed as hazard ratios (HRs) calculated using a Cox proportional hazard regression model indicating the multiplicative change in incidence rate (hazard) per 1.1 mmol/mol or (0.1% point) increment in HbA$_{1c}$. The effect of the yearly HbA$_{1c}$ change (the slope of the regression line) on the outcome was expressed as HRs per 1.1 mmol/mol or (0.1% point) increase in HbA$_{1c}$/year. The effect of the 1-year HbA$_{1c}$ level and yearly change in HbA$_{1c}$ was assessed in unadjusted as well as in adjusted models: Model 1 included age, sex, living alone, BMI (and one-year HbA$_{1c}$ levels in the analyses including change); Model 2 additionally included hypertension, total cholesterol, triglycerides, LTPA, smoking status, and wasting. To assess the linearity of the effect of glycaemic control, a restricted cubic spline was used to model the effect of glycaemic control after adjustment as in Model 2. Sensitivity analyses were performed adjusted for urinary albumin, creatinine, and anti-diabetic medication (Model 3). To investigate if one-year HbA$_{1c}$ level, sex, and anti-diabetic medication affected the relation between the yearly HbA$_{1c}$ change and risk of morbidity and mortality, we repeated the aforementioned analyses, stratified on one-year HbA$_{1c}$ level, sex, and type of treatment. All analyses were performed in SAS (Version 9.4).

To validate the linearity assumptions of our general linear regression models, we performed non-linear spline regression analyses on the associations between yearly HbA$_{1c}$ change and the clinical outcomes. The incidence of microvascular disease increased approximately linearly with increasing slope of the regression line (data not shown).

## Results

The median age was 69.1 year at the end of the intervention, on average six years after diagnosis, and 50% of participants were men (Table 1). At the end of follow-up, 62% received oral anti-diabetic medication and 29% was treated with diet only. One year after diagnosis, HbA1c levels were on average lower than at time of diagnosis. During the following five years the observed average HbA$_{1c}$ increased with 0.19% per year (95% confidence interval (95% CI): 0.01–0.43) (Table 1), (equivalent to a rise of. 2.1 mmol/mol (95% CI: 0.11–4.70)).

Higher HbA$_{1c}$ levels one year after diagnosis were associated with a higher risk of later diabetes-related morbidity and mortality (all p value < 0.05; see Table 2).

Next we assessed the relation between yearly HbA$_{1c}$ changes dichotomized on the median (0.188 percentage point per year during the 6-year intervention period) and the incidence of various outcomes during the 13 years thereafter. Table 3 shows the risk for various outcomes for those with a median or higher increase in HbA1 levels during the intervention period, relative to those with change in HbA1c below the median. An increase in HbA1C levels equal or higher than median was associated with a higher risk of myocardial infarction during the follow-up.

**Table 1. Participant characteristics.**

| | n | Characteristics |
|---|---|---|
| Baseline HbA$_{1c}$ level (mmol/mol) [1] | 411 | 88 (72–103) |
| Baseline HbA$_{1c}$ level (%) [1] | | 10.2 (8.7–11.6) |
| One-year HbA$_{1c}$ level (mmol/mol) [1] | 494 | 60 (52–71) |
| One-year HbA$_{1c}$ level (%) [1] | | 7.65 (6.91–8.62) |
| Yearly HbA$_{1c}$ change (mmol/mol/year) | 494 | 2.1 (0.11–4.70) |
| Yearly HbA$_{1c}$ change (%/ year) | | 0.19 (0.01–0.43) |
| Age at end study (years) | 494 | 69.1 (60.1–77.4) |
| Sex (male) (n,%) | 494 | 244 (49.4) |
| Living alone (n,%) | 459 | 167 (36.4) |
| BMI (kg/m2) | 483 | 28.3 (25.6–31.7) |
| Baseline systolic blood pressure (mmHg) | 489 | 150 (132–160) |
| Baseline diastolic blood pressure (mmHg) | 489 | 85 (80–90) |
| Systolic blood pressure (mmHg) | 492 | 149 (132–160) |
| Diastolic blood pressure (mmHg) | 492 | 80 (80–90) |
| Hypertension [2] (n,%) | 494 | 361 (73.1) |
| Total cholesterol (mmol/l) | 491 | 6.0 (5.2–6.8) |
| Fasting triglycerides (mmol/l) | 463 | 1.73 (1.22–2.50) |
| Sedentary leisure time physical activity | 453 | 128 (28.3) |
| Smoking (n,%) | 457 | 146 (32.0) |
| Wasting [3] (n,%) | 494 | 82 (16.6) |
| Anti-diabetic treatment (n,%) | 494 | |
| Diet only | | 142 (28.7) |
| Oral agents | | 304 (61.5) |
| Insulin | | 48 (9.7) |
| Serum creatinine (μmol/l) | 491 | 89 (80–103) |
| Urinary albumin (n,%) | 468 | |
| Normal | | 290 (62.0) |
| Microalbuminuria | | 158 (33.8) |
| Proteinuria | | 20 (4.3) |

Unless otherwise stated, participant characteristics are from six years after diabetes diagnosis. Data are numbers (%) or medians (interquartile range (IQR)). N denotes the number of participants for whom variables were available.
[1]Reference value 36–57 mmol/mol (5.4–7.4%);
[2]Hypertension: systolic/diastolic blood pressure $\geq$ 160 and/or 90 mmHg;
[3]Wasting: unintentional weight loss $\geq$ 1 kg per year in participants without intention to lose weight.

Table 4 shows the relation between continuous changes in HbA1c levels during the intervention period Change in HbA$_{1c}$ during the intervention period was not associated with the aggregate outcome any diabetes-related endpoint, all-cause mortality, diabetes-related mortality, myocardial infarction, and stroke in the multivariable analyses. However, yearly HbA$_{1c}$ change was associated with a higher incidence of microvascular complications (HR, 95% CI: 1.14, 1.05–1.24) and peripheral vascular disease (1.14, 1.00–1.30). In sensitivity analyses, when additionally adjusting for urinary albumin, creatinine, and anti-diabetic medication, the association between yearly HbA$_{1c}$ change and microvascular disease remained (1.11; 1.01–1.21; $P = 0.024$), but the association with peripheral vascular disease was attenuated (1.05; 0.89–1.24; $P = 0.58$).

**Table 2. Relation between one-year level of haemoglobin A$_{1c}$ and risk of outcomes during 13 years of post-intervention follow-up.**

| | Events before or during intervention (n, %) | Events during post-intervention follow-up (n, %) | Model 1 | | Model 2 | | Model 3 | |
|---|---|---|---|---|---|---|---|---|
| | | | HR (95% CI) | P value | HR (95% CI) | P value | HR (95% CI) | P value |
| Any diabetes-related endpoint | 157 (31.9) | 196 (58.3) | 1.24 (1.09–1.41) | 0.001 | 1.26 (1.10–1.45) | 0.001 | 1.19 (1.01–1.41) | 0.03 |
| All-cause mortality | - | 295 (59.7) | 1.14 (1.02–1.27) | 0.02 | 1.18 (1.05–1.33) | 0.006 | 1.14 (0.98–1.33) | 0.09 |
| Diabetes-related mortality | - | 186 (37.8) | 1.18 (1.02–1.35) | 0.02 | 1.24 (1.07–1.44) | 0.005 | 1.3 (1.08–1.55) | 0.005 |
| Myocardial infarction | 46 (9.4) | 121 (27.1) | 1.16 (0.96–1.38) | 0.09 | 1.22 (1.01–1.48) | 0.04 | 1.34 (1.05–1.70) | 0.02 |
| Stroke | 47 (9.5) | 78 (17.5) | 1.42 (1.12–1.79) | 0.004 | 1.46 (1.13–1.88) | 0.004 | 1.37 (0.98–1.91) | 0.07 |
| Peripheral vascular disease | 2 (0.4) | 20 (4.1) | 2.05 (1.44–2.91) | <0.0001 | 1.85 (1.28–2.68) | 0.001 | 1.43 (0.93–2.21) | 0.10 |
| Microvascular disease | 15 (3.1) | 54 (11.3) | 1.55 (1.24–1.92) | 0.0001 | 1.68 (1.33–2.13) | <0.0001 | 1.48 (1.12–1.96) | 0.006 |

Events are given as count data with percentages (%). Other data are given as hazard ratios (HR) with 95% confidence interval (CI) and corresponding p values for outcomes during 13 years of post-intervention follow-up expressed per 1.1 mmol/mol (0.1%) increment in levels of HbA$_{1c}$ at 1 year after diagnosis. Model 1 is adjusted for age, sex, living alone, and BMI. Model 2: as model 1, but additionally adjusted for hypertension, cholesterol, triglycerides, sedentary leisure time physical activity, smoking, and wasting. Model 3: as model 2, but additionally adjusted for anti-diabetic medication, urinary albumin, and creatinine.

In additional analyses, we investigated whether the effect of yearly HbA$_{1c}$ change on the outcomes was dependent on the participant's characteristics, such as one-year HbA$_{1c}$ level, sex, and type of anti-diabetic treatment (Table 5). The relation between yearly HbA$_{1c}$ change and incidence of all-cause and diabetes-related mortality was stronger for women than for

**Table 3. Relation between dichotomized change in haemoglobin A$_{1c}$ during the 6-year intervention period and risk of outcomes during the subsequent 13 years of follow-up.**

| | Events before or during intervention (n, %) | Events during post-intervention follow-up (n, %) | Model 1 | | Model 2 | | Model 3 | |
|---|---|---|---|---|---|---|---|---|
| | | | HR (95% CI) | P value | HR (95% CI) | P value | HR (95% CI) | P value |
| Any diabetes-related endpoint | 157 (31.9) | 196 (58.3) | 0.91 (0.65–1.28) | 0.59 | 0.92 (0.63–1.32) | 0.63 | 0.82 (0.55–1.22) | 0.33 |
| All-cause mortality | - | 295 (59.7) | 1.22 (0.94–1.60) | 0.14 | 1.24 (0.94–1.64) | 0.13 | 1.12 (0.83–1.50) | 0.46 |
| Diabetes-related mortality | - | 186 (37.8) | 1.40 (0.99–1.98) | 0.060 | 1.40 (0.96–2.03) | 0.077 | 1.27 (0.86–1.86) | 0.23 |
| Myocardial infarction | 46 (9.4) | 121 (27.1) | 1.81 (1.17–2.79) | 0.0074 | 2.14 (1.30–3.52) | 0.0030 | 2.07 (1.26–3.41) | 0.0040 |
| Stroke | 47 (9.5) | 78 (17.5) | 0.95 (0.52–1.73) | 0.86 | 1.01 (0.51–2.02) | 0.98 | 0.89 (0.43–1.85) | 0.76 |
| Peripheral vascular disease | 2 (0.4) | 20 (4.1) | 2.09 (0.70–6.19) | 0.19 | 1.47 (0.55–3.95) | 0.45 | 1.02 (0.40–2.63) | 0.97 |
| Microvascular disease | 15 (3.1) | 54 (11.3) | 1.67 (0.85–3.27) | 0.13 | 1.71 (0.85–3.47) | 0.14 | 1.49 (0.74–3.05) | 0.27 |

Events are given as count data with percentages (%). Other data are given as hazard ratios (HR) with 95% confidence interval (CI). Hazard ratios denote the effect of HbA1c change equal and above median relative to HbA1c change below the median during the 6-year intervention period after diagnosis on outcomes during 13 years of post-intervention. Model 1 is adjusted for age, sex, living alone, BMI, and one-year HbA$_{1c}$ level. Model 2: as model 1, but additionally adjusted for hypertension, cholesterol, triglycerides, sedentary leisure time physical activity, smoking, and wasting. Model 3: as model 2, but additionally adjusted for anti-diabetic medication, urinary albumin, and creatinine.

**Table 4. Relation between yearly change in haemoglobin A$_{1c}$ during the 6-year intervention period and risk of outcomes during the subsequent 13 years of follow-up.**

| | Events before or during intervention (n, %) | Events during post-intervention follow-up (n, %) | Model 1 | | Model 2 | | Model 3 | |
|---|---|---|---|---|---|---|---|---|
| | | | HR (95% CI) | P value | HR (95% CI) | P value | HR (95% CI) | P value |
| Any diabetes-related endpoint | 157 (31.9) | 196 (58.3) | 0.99 (0.95–1.04) | 0.71 | 1 (0.95–1.05) | 0.94 | 0.98 (0.92–1.03) | 0.42 |
| All-cause mortality | - | 295 (59.7) | 1.01 (0.97–1.05) | 0.66 | 1.02 (0.98–1.07) | 0.32 | 1.00 (0.96–1.06) | 0.82 |
| Diabetes-related mortality | - | 186 (37.8) | 1.00 (0.96–1.05) | 0.86 | 1.01 (0.96–1.07) | 0.63 | 1.00 (0.94–1.07) | 0.96 |
| Myocardial infarction | 46 (9.4) | 121 (27.1) | 1.02 (0.96–1.08) | 0.57 | 1.04 (0.97–1.11) | 0.23 | 1.06 (0.99–1.14) | 0.10 |
| Stroke | 47 (9.5) | 78 (17.5) | 1.02 (0.94–1.10) | 0.65 | 1.04 (0.95–1.14) | 0.40 | 1.01 (0.91–1.13) | 0.81 |
| Peripheral vascular disease | 2 (0.4) | 20 (4.1) | 1.17 (1.05–1.30) | 0.006 | 1.14 (1.00–1.30) | 0.047 | 1.05 (0.89–1.24) | 0.59 |
| Microvascular disease | 15 (3.1) | 54 (11.3) | 1.11 (1.03–1.20) | 0.006 | 1.14 (1.05–1.24) | 0.002 | 1.11 (1.01–1.21) | 0.024 |

Events are given as count data with percentages (%). Other data are given as hazard ratios (HR) with 95% confidence interval (CI) and corresponding p values for outcomes during 13 years of post-intervention follow-up expressed per 1.1 mmol/mol (0.1%) per year change in HbA$_{1c}$ during the 6-year intervention period after diagnosis. Model 1 is adjusted for age, sex, living alone, BMI, and one-year HbA$_{1c}$ level. Model 2: as model 1, but additionally adjusted for hypertension, cholesterol, triglycerides, sedentary leisure time physical activity, smoking, and wasting. Model 3: as model 2, but additionally adjusted for anti-diabetic medication, urinary albumin, and creatinine.

men (p for interaction 0.033 and 0.038 respectively). Regarding the modifying effect of diabetic medication, treatment with insulin was associated with an increased risk of diabetes-related mortality compared to participants treated with oral medication or diet only. Moreover, oral medication was a risk indicator for later myocardial infarction when compared to diet only and insulin therapy.

## Discussion

In this Danish population-based sample of persons with newly diagnosed Type 2 DM, a higher yearly increase in HbA$_{1c}$ was an independent risk factor for microvascular disease, but not for macrovascular disease or mortality during the subsequent 13-years in participants receiving structured personal care in general practice. The relation between yearly HbA$_{1c}$ change and the incidence of microvascular outcomes appeared to be linear. Although increasing HbA$_{1c}$ was associated with a higher incidence of peripheral vascular disease, this association was no longer significant after correction for urinary albumin, creatinine, and anti-diabetic medication. These findings were independent of the level of HbA$_{1c}$ after one year of treatment. We found no association between an increase in HbA$_{1c}$ during the intervention period and mortality or incident cardiovascular events during 13 years of follow-up. In the DCGP trial the intervention reduced the incidence of myocardial infarction after 19 years of follow-up compared with the control group (9). This time point coincides with the end of the 13-year follow-up in the present study. Thus, the present observational study suggest that the reduction in macrovascular disease may not attributable to quality of HbA$_{1c}$ control during the intervention period. but may bedue to other elements of the intervention like blood pressure control. Alternatively, the lower macrovascular morbidity in the DCGP study at 19 years may be the results of the legacy effect, the long-term health benefits of early glycemic control.

**Table 5. Modifying effect of one-year HbA$_{1c}$ level, sex, and antidiabetic treatment, on the relation between yearly change in in haemoglobin A$_{1c}$ during the 6-year intervention period and risk for various outcomes during the subsequent 13-year post-intervention follow-up.**

| | Effect modifier | Events before or during intervention (n, %) | Events during post-intervention follow-up (n, %) | HR (95% CI) | P value | P for interaction |
|---|---|---|---|---|---|---|
| | | *HbA$_{1c}$ level, mmol/mol (%)* | | | | |
| Any diabetes-related endpoint | ≤57.4 (7.4) | 59 (28.5) | 74 (50.0) | 0.95 (0.87–1.04) | 0.24 | 0.14 |
| | >57.4 (7.4) | 98 (34.3) | 122 (64.9) | 1.01 (0.96–1.06) | 0.77 | |
| All-cause mortality | ≤57.4 (7.4) | 0 | 118 (57.0) | 1.04 (0.98–1.11) | 0.22 | 0.49 |
| | >57.4 (7.4) | 0 | 177 (61.7) | 1.01 (0.97–1.06) | 0.58 | |
| Diabetes-related mortality | ≤57.4 (7.4) | 0 | 72 (34.8) | 1.01 (0.93–1.10) | 0.78 | 0.97 |
| | >57.4 (7.4) | 0 | 114 (40.0) | 1.01 (0.95–1.08) | 0.66 | |
| Myocardial infarction | ≤57.4 (7.4) | 18 (8.7) | 49 (25.2) | 1.07 (0.97–1.18) | 0.15 | 0.46 |
| | >57.4 (7.4) | 29 (9.8) | 72 (28.0) | 1.03 (0.95–1.11) | 0.45 | |
| Stroke | ≤57.4 (7.4) | 17 (8.2) | 23 (12.1) | 0.93 (0.77–1.13) | 0.47 | 0.16 |
| | >57.4 (7.4) | 30 (10.5) | 55 (21.5) | 1.05 (0.96–1.15) | 0.25 | |
| Peripheral vascular disease | ≤57.4 (7.4) | 1 (0.5) | 4 (1.9) | 1.14 (0.92–1.41) | 0.24 | 0.99 |
| | >57.4 (7.4) | 1(0.4) | 16 (5.6) | 1.14 (1.00–1.30) | 0.048 | |
| Microvascular disease | ≤57.4 (7.4) | 5 (2.4) | 15 (7.4) | 1.10 (0.95–1.27) | 0.21 | 0.49 |
| | >57.4 (7.4) | 10 (3.5) | 39 (14.2) | 1.15 (1.06–1.25) | 0.001 | |
| | *Sex* | | | | | |
| Any diabetes related endpoint | Women | 78 (31.3) | 103 (60.2) | 1.02 (0.96–1.09) | 0.47 | 0.18 |
| | Men | 79 (32.4) | 93 (56.4) | 0.98 (0.92–1.03) | 0.41 | |
| All-cause mortality | Women | 0 | 133 (53.2) | 1.07 (1.01–1.13) | 0.017 | 0.033 |
| | Men | 0 | 162 (66.4) | 0.99 (0.94–1.04) | 0.67 | |
| Diabetes related mortality | Women | 0 | 86 (34.5) | 1.07 (1.00–1.15) | 0.049 | 0.038 |
| | Men | 0 | 100 (41.2) | 0.97 (0.91–1.04) | 0.46 | |
| Myocardial infarction | Women | 19 (7.6) | 55 (23.9) | 1.09 (1.01–1.18) | 0.027 | 0.15 |
| | Men | 27 (11.1) | 66 (30.9) | 1.01 (0.92–1.10) | 0.89 | |
| Stroke | Women | 22 (8.8) | 35 (15.4) | 1.08 (0.96–1.21) | 0.20 | 0.35 |
| | Men | 25 (10.3) | 43 (19.6) | 1.02 (0.92–1.13) | 0.72 | |
| Peripheral vascular disease | Women | 1 (0.4) | 5 (2.0) | 1.01 (0.74–1.37) | 0.95 | 0.32 |

*(Continued)*

**Table 5.** (Continued)

| | Effect modifier | Events before or during intervention (n, %) | Events during post-intervention follow-up (n, %) | HR (95% CI) | P value | P for interaction |
|---|---|---|---|---|---|---|
| | Men | 1 (0.4) | 15 (6.2) | 1.17 (1.03–1.33) | 0.012 | |
| Microvascular disease | Women | 6 (2.4) | 26 (10.7) | 1.15 (1.03–1.28) | 0.010 | 0.78 |
| | Men | 9 (3.7) | 28 (12.0) | 1.13 (1.02–1.25) | 0.017 | |
| | *Antidiabetic treatment* | | | | | |
| Any diabetes-related endpoint | Diet only | 42 (29.6) | 50 (50.0) | 0.91 (0.84–1.00) | 0.046 | 0.030 |
| | Oral | 99 (32.7) | 124 (60.8) | 1.02 (0.96–1.09) | 0.43 | |
| | Insulin | 16 (33.3) | 22 (68.8) | 0.93 (0.83–1.04) | 0.22 | |
| All-cause mortality | Diet only | 0 | 84 (59.2) | 0.93 (0.86–1.01) | 0.074 | 0.021 |
| | Oral | 0 | 181 (59.5) | 1.04 (0.99–1.10) | 0.11 | |
| | Insulin | 0 | 30 (62.5) | 1.04 (0.96–1.12) | 0.33 | |
| Diabetes-related mortality | Diet only | 0 | 60 (42.3) | 0.95 (0.86–1.04) | 0.26 | 0.023 |
| | Oral | 0 | 105 (34.8) | 1.04 (0.97–1.11) | 0.29 | |
| | Insulin | 0 | 21 (43.8) | 1.11 (1.01–1.22) | 0.030 | |
| Myocardial infarction | Diet only | 14 (9.9) | 36 (28.1) | 0.95 (0.86–1.06) | 0.36 | 0.036 |
| | Oral | 28 (9.3) | 74 (27.0) | 1.10 (1.02–1.19) | 0.019 | |
| | Insulin | 4 (8.3) | 11 (25.0) | 1.05 (0.93–1.19) | 0.46 | |
| Stroke | Diet only | 19 (13.4) | 19 (15.5) | 0.97 (0.84–1.13) | 0.71 | 0.57 |
| | Oral | 23 (7.6) | 47 (16.8) | 1.05 (0.94–1.17) | 0.37 | |
| | Insulin | 5 (10.4) | 12 (27.9) | 1.04 (0.89–1.21) | 0.66 | |
| Peripheral vascular disease | Diet only | 0 (0.0) | 1 (0.7) | 1.48 (1.17–1.86) | 0.001 | 0.053 |
| | Oral | 2 (0.7) | 14 (4.7) | 1.07 (0.90–1.28) | 0.45 | |
| | Insulin | 0 (0.0) | 5 (10.4) | 1.12 (0.90–1.39) | 0.30 | |
| Microvascular disease | Diet only | 4 (2.8) | 6 (4.4) | 1.03 (0.90–1.19) | 0.67 | 0.29 |
| | Oral | 10 (3.3) | 41 (14.0) | 1.15 (1.05–1.26) | 0.002 | |
| | Insulin | 1 (2.1) | 7 (14.9) | 1.06 (0.83–1.36) | 0.64 | |

Events are given as count data (n) with percentages (%). Other data are given as hazard ratios (HR) with 95% confidence interval (CI) and corresponding p values for outcomes during 13 year post-intervention follow-up expressed per 1.1 mmol/mol (0.1%) per year change in HbA$_{1c}$ during 6-year intervention period after diagnosis adjusted for age, sex, living alone, BMI, hypertension, total cholesterols, triglycerides, leisure time physical activity, smoking, and wasting.

A recent study investigated the relation between the difference in HbA$_{1c}$ levels before and after 6 months of treatment with metformin and the subsequent risk of cardiovascular events and mortality. This study found that a decrease in HbA$_{1c}$ during 6 months of treatment was associated with a lower risk of cardiovascular events and mortality during a median follow-up of 2.8 years in participants [17]. Admittedly, the statistical precision of these results was limited and, since all participants used metformin, it is possible that the effect was due to the pharmacological properties of metformin instead of the decrease in HbA$_{1c}$. Yet in line with these earlier findings, we also found that increasing HbA$_{1c}$ levels during the six-year intervention period were associated with an increased risk of microvascular complications and a tendency towards higher risk of peripheral vascular morbidity. Increasing levels of HbA$_{1c}$ during the intervention period may have been caused by a less intensive treatment of diabetes, or may be ascribed to diabetes subtype, weight gain, or other factors. A meta-analysis demonstrated that intensive treatment of Type 2 DM lowers the risk of microvascular complications as well as the risk of amputation of lower extremities. [18]. However, this meta-analysis failed to show an effect on macro-vascular morbidity. A follow-up study of the Veteran Affair Diabetes Trial (VADT) concluded that intensive treatment may reduce the risk of major cardiovascular events, but that the intervention did not lower mortality risk [19]. Thus, overall taken, results regarding the risk of increasing HbA$_{1c}$ levels are not consistent between studies. This may in part be due to methodological heterogeneity, e.g. differences in treatment regimens, duration of follow-up, and definition of quality of glycaemic control as well as participant characteristics and compliance issues.

The UKPDS study is comparable to the DCGP study, in that it included persons with newly diagnosed Type 2 DM, while the ADVANCE and ACCORD studies included persons already diagnosed with diabetes with comorbid cardiovascular disease and a mean diabetes duration of eight and ten years, respectively [20, 21]. In UKPDS, the measure of HbA$_{1c}$ was an updated mean of HbA$_{1c}$ based on the mean of the annual measurements during follow-up, which mathematically is essentially the same analysis we conducted for the current study. The UKPDS found that a 1% lower updated mean HbA$_{1c}$ was associated with a 37% lower risk for microvascular complications, and a 43% lower risk of the combined endpoint of amputation or death by peripheral vascular disease [11]. These results by and large concord with our findings. However, unlike the UKPDS study, which reported a lower all-cause and diabetes-related mortality of respectively 14% and 21% per 1% lower updated mean HbA$_{1c}$ [11, 21], we only found an association between HbA$_{1c}$ and mortality in women. A sex-specific difference in diabetes outcomes is reminiscent of earlier findings in the DCGP study of a post-hoc analysis between structured personal care compared with routine care [12]. Women receiving structured personal care showed a lower all-cause mortality, incidence of diabetes-related mortality, and any diabetes-related endpoint than women receiving routine care, while this difference was absent in men. This sex-related difference was ascribed to complex social and cultural issues involving gender [12]. Regarding microvascular complications, the UKPDS concluded that the rate of increase of the risk for microvascular disease with hyperglycaemia was larger than that for macrovascular disease [11].

A major strength of this study is the prospective design and the long-term post-trial follow up. Use of Danish registry data ensured that follow-up was virtually complete [10]. The included sample is likely to be representative of community-dwelling persons with Type 2 DM that have survived for some years with this diagnosis and have followed a structured disease management program, which nowadays is included in the clinical guidelines for diabetes. The change in HbA$_{1c}$ over time is a highly clinically relevant parameter since decisions on treatment adjustments are typically based on HbA$_{1c}$ measurements. The inclusion of repeated

HbA$_{1c}$ measurements is anticipated to take into account a considerable proportion of the variation in the disease progression in persons with Type 2 DM [15].

Microvascular disease was treated as a surrogate measure of disease status. The outcome is a composite outcome including renal complications and eye-related complications, and the effect sizes in our study were expectedly attenuated when results were adjusted for urinary albumin and antidiabetic treatment.

Our study has a number of shortcomings. The analyses in this study are based on information of the disease progression during the intervention of the trial, and no information was available on treatment or HbA$_{1c}$ levels during the succeeding 13 years of follow-up. However, it has been shown that although glucose regulation deteriorates in treatment groups after the end of the intervention, a beneficial "legacy effect" on cardiovascular end-points may persist [22] [23]. A second shortcoming is large number of significance tests performed. We had no predefined ideas on how changes in HbA1c would affect later diabetes-related outcomes. The study is therefore hypothesis generating, and the found associations should therefore be confirmed in subsequent, hypothesis confirming studies. This implicates though that the typical p-value threshold of 5% should be interpreted with caution.

## Conclusions

In conclusion, increasing HbA$_{1c}$ levels during the first six years after the diagnosis of Type 2 DM were associated with a higher incidence of microvascular complications, but not macrovascular complications and mortality during the following 13 years. These findings should further encourage clinicians to pursue optimal glycaemic control to prevent long-term microvascular complications in persons with Type 2 DM.

## Supporting information

**S1 Table. Definition of clinical outcomes in the Diabetes Care in General Practice (DCGP) 19-year registry-based monitoring used to classify cause of death or morbidity.** [a] The Danish National Death Registry and the National Hospital Discharge Registry changed coding from International Classification of Diseases 8 (ICD-8) to ICD-10 on 1 January 1994. The Danish National Death Registry contains only the first 4 characters of the ICD codes, while the National Hospital Discharge Registry contains all 5 characters. [b] The National Hospital Discharge Registry changed coding of surgical procedures from the 3rd edition of The Danish Classification of Surgical Procedures to the Nordic Classification of Surgical Procedures on 1 January 1996.
(DOCX)

## Acknowledgments

We thank Niels de Fine Olivarius from the Research Unit for General Practice, Department of Public Health, University of Copenhagen, Denmark, for his contribution to this study.

## Author Contributions

**Conceptualization:** Anne Møller, Rune Aabenhus, Volkert Siersma, Katja Rasmussen, Rasmus Køster-Rasmussen.

**Formal analysis:** Maarten P. Rozing, Volkert Siersma.

**Writing – original draft:** Anne Møller, Rune Aabenhus, Rasmus Køster-Rasmussen.

**Writing – review & editing:** Maarten P. Rozing, Anne Møller, Rune Aabenhus, Volkert Siersma, Katja Rasmussen.

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
