## [Decision Letter · Decision Letter 0]

31 Jul 2019

PONE-D-19-16655

Changes in HbA1c during the first six years after the diagnosis of Type 2 diabetes mellitus predict long-term microvascular outcomes

PLOS ONE

Dear Dr Rozing,

Thank you for submitting your manuscript to PLOS ONE. After careful consideration, we feel that it has merit but does not fully meet PLOS ONE’s publication criteria as it currently stands. Therefore, we invite you to submit a revised version of the manuscript that addresses the points raised during the review process.

We would appreciate receiving your revised manuscript by Sep 14 2019 11:59PM. To enhance the reproducibility of your results, we recommend that if applicable you deposit your laboratory protocols in protocols.io, where a protocol can be assigned its own identifier (DOI) such that it can be cited independently in the future. For instructions see: http://journals.plos.org/plosone/s/submission-guidelines#loc-laboratory-protocols

We look forward to receiving your revised manuscript.

Kind regards,

Tatsuo Shimosawa, M.D., Ph.D.

Academic Editor

PLOS ONE

Journal Requirements:

2. In your ethics statement in the manuscript and in the online submission form, please provide additional information about the patient records used in your retrospective study. Specifically, please ensure that you have discussed whether all data were fully anonymized before you accessed them and/or whether the IRB or ethics committee waived the requirement for informed consent. If patients provided informed written consent to have data from their medical records used in research, please include this information.

4. Thank you for stating the following in the Financial Disclosure section:

[Major funding for this study was received from The Danish Medical Research Council, The Danish Research Foundation for General Practice, The Health Insurance Foundation, The Danish Ministry of Health, Novo Nordisk Farmaka Denmark Ltd., and The Pharmacy Foundation. The funders played no role in the design, conduct, analysis, interpretation of data, or reporting of the study.].

We note that you received funding from a commercial source: [Novo Nordisk Farmaka Denmark Ltd]

Reviewers' comments:

Reviewer's Responses to Questions

**Comments to the Author**

1. Is the manuscript technically sound, and do the data support the conclusions?

Reviewer #1: Partly

Reviewer #2: Partly

2. Has the statistical analysis been performed appropriately and rigorously? 

Reviewer #1: I Don't Know

Reviewer #2: I Don't Know

3. Have the authors made all data underlying the findings in their manuscript fully available?

Reviewer #1: No

Reviewer #2: No

4. Is the manuscript presented in an intelligible fashion and written in standard English?

Reviewer #1: Yes

Reviewer #2: Yes

5. Review Comments to the Author

Reviewer #1: It has already been well-known that higher level of HbA1c is significantly associated with diabetic (especially, microvascular) complications. The observational period and number of subjects in this study are too few to get any conclusive results. Therefore, there is virtually very few novelty anyway. Trajectory of HbA1c is also really depend upon inclusion criteria and backgrounds of the patients, so that extrapolation of their results and their clinical relevance are also very limited.

Reviewer #2: Rozing et al examined the impact of annual changes of HbA1c during intervention period in DCGP study on diabetes-related outcomes. In general, although many of the data is confirmatory, it is still of importance with regard to regular clinical care for newly diagnosed diabetes. However, there are several points to be addressed to improve the manuscript, and to make it more comprehensive.

1. The authors stated that one year HbA1c level during intervention period strongly predicts macro- and microvascular diabetic outcomes (supplemental table 1), although the data is confirmatory of their previous publication (ref 9) as understood as “legacy effect”. In contrast, impacts of annual changes in HbA1c after 1-year treatment appear not so impressive. The reviewer is wondering what the result would be, if the annual changes in HbA1c after 1-year treatment is categorized into 2 categories, such as maintained and worse based on the median. If this categorization predicts major diabetic outcomes, it highlights the significance of keeping glycemic control for several years following newly onset of diabetes.

2. Supplemental figure 2 should be mistyping, which should be indeed figure 2. This figure is not easily understood for regular clinicians. The reviewer cannot judge whether this figure is appropriately analyzed and demonstrated. This should be reviewed by an appropriate biological statistician selected by the Journal.

3. The reviewer feels that all supplemental data should be incorporated into the main text as regular Figures and Tables. Is the format including references is in line with the PLoS ONE? It should be carefully checked by the authors and the editorial technical staff.

4. page 3, line 3: The sentence of “anti-diabetic medication lowers the risk of complications (ref 6)” is not appropriate, since the research compares 2 types of anti-diabetic medications.

5. page 4, line 18: Does this mean that patients with the use of insulin within 180 days (2.5%) were not regarded as type 2 diabetes? This classification is clearly not appropriate.

6. page 5, line 2: Is the sentence “honey moon phase” properly used?

7. page 5, definition of outcomes line 1: Please clarify Diabetes-related endpoint and Diabetes-related mortality in this Section, although they are listed in Supplemental Table 2.

8. page 7, line 5: What does it mean by “HbA1c was, on average, normalized”? Please clarify.

9. page 8, line 5, Supplemental table 2: The authors only stated the impact of insulin therapy on diabetes-related death. The effect of oral therapy on myocardial infarction is also significant, with significant interaction. This should also be described and discussed.

10. page 8, last line: The reviewer is against the idea that “the reduction in macrovascular disease was not attributable to poor HbA1c control,” since this may be the result of “legacy effect”. Please clarify.

11. page 9, line 14, ref 19: This reference dealt with microvascular complications besides amputation. Both of these findings should be discussed.

6. PLOS authors have the option to publish the peer review history of their article (what does this mean?). If published, this will include your full peer review and any attached files.

Reviewer #1: No

Reviewer #2: No

---

## [Author Response · Author response to Decision Letter 0]

8 Oct 2019

Dear editor,

We thank you and two anonymous reviewers for the comments and suggestions regarding the previous version of our manuscript Changes in HbA1c during the first six years after the diagnosis of Type 2 diabetes mellitus predict long-term microvascular outcomes (ONE-D-19-16655). We have revised the manuscript accordingly. Our point-to-point responses to the comments are given below and are marked with an hash sign (#). Any changes in the text are marked with an asterisk (*). The letter “l” followed by a number, denotes the line number where changes in the text were made in the marked version of the manuscript.

Comments to the Author

5. Review Comments to the Author

1. Reviewer #1: It has already been well-known that higher level of HbA1c is significantly associated with diabetic (especially, microvascular) complications. 

# We agree that this is a confirmatory rather than an exploratory study. We make this more explicit in the current version of our manuscript:

*“In the present confirmatory study, the aim was to determine the relation between changes in HbA1c during 6 years of intervention in persons with newly diagnosed Type 2 DM and any diabetes-related endpoint, all-cause mortality, diabetes-related mortality, myocardial infarction, stroke, peripheral vascular disease and microvascular disease during the 13 years thereafter.” (l. 49)

2. The observational period and number of subjects in this study are too few to get any conclusive results. Therefore, there is virtually very few novelty anyway. 

#The intervention period and the follow-up period encompassed 6 and 13 years respectively. When studying longer-term complications of diabetes in humans this time-frame should be sufficient. Our sample included 494 patients of which approximately 60% experienced a diabetes-related event. A larger sample size would have allowed for higher accuracy, and would merely have yielded more statistically significant results, yet not necessarily more clinically significant results. 

3. Trajectory of HbA1c is also really depend upon inclusion criteria and backgrounds of the patients, so that extrapolation of their results and their clinical relevance are also very limited.

#We fully agree with your concern regarding external validity when studying HbA1c trajectories. Especially in randomized controlled trials, preferential inclusion of distinct patient characteristics limits the applicability of their findings to the general population. However, in our study with virtually no exclusion criteria, the study sample represents a population, which clinicians typically encounter in their everyday clinical practice.

Reviewer #2: Rozing et al examined the impact of annual changes of HbA1c during intervention period in DCGP study on diabetes-related outcomes. In general, although many of the data is confirmatory, it is still of importance with regard to regular clinical care for newly diagnosed diabetes. However, there are several points to be addressed to improve the manuscript, and to make it more comprehensive.

1. The authors stated that one year HbA1c level during intervention period strongly predicts macro- and microvascular diabetic outcomes (supplemental table 1), although the data is confirmatory of their previous publication (ref 9) as understood as “legacy effect”. In contrast, impacts of annual changes in HbA1c after 1-year treatment appear not so impressive. The reviewer is wondering what the result would be, if the annual changes in HbA1c after 1-year treatment is categorized into 2 categories, such as maintained and worse based on the median. If this categorization predicts major diabetic outcomes, it highlights the significance of keeping glycemic control for several years following newly onset of diabetes.

#Thank you for your suggestion. We included an additional Table 3 accordingly, showing the results for those with “worsened HbA1c” levels and “stable/improved HbA1c” levels. We included the following sentence in the results section:

*“Next we assessed the relation between yearly HbA1c changes during the intervention period and the incidence of various outcomes during the 13 years thereafter. Table 3 shows the risk for various outcomes for those with a median (0.188 percentage point per year during the 6-year intervention period) or higher increase in HbA1 levels during the intervention period, relative to those with change in HbA1c below the median. An increase in HbA1C levels equal or higher than median was associated with a higher risk of myocardial infarction during the follow-up.” (l.149-155)

2. Supplemental figure 2 should be mistyping, which should be indeed figure 2. This figure is not easily understood for regular clinicians. The reviewer cannot judge whether this figure is appropriately analyzed and demonstrated. This should be reviewed by an appropriate biological statistician selected by the Journal.

#We agree that interpretation of the figure is cumbersome and may therefore not be informative to regular clinicians. The main message of the figure was to show the linear relationship between the change (“slope”) of the 1%/year increment in hemoglobin A1c during the 6-year intervention period and the risk microvascular disease during 13-year post-intervention follow-up. We removed the figure from the current version of the manuscript and moved the following sentence from the results section to the statistics section: 

*“To validate the linearity assumptions of our general linear regression models, we performed non-linear spline regression analyses on the associations between yearly HbA1c change and the clinical outcomes. The incidence of microvascular disease increased approximately linearly with increasing slope of the regression line (data not shown).” (l.133-136)

3. The reviewer feels that all supplemental data should be incorporated into the main text as regular Figures and Tables. Is the format including references is in line with the PLoS ONE? It should be carefully checked by the authors and the editorial technical staff.

#*We incorporated all supplemental tables and figures in the main text, except for table S1 (previously table S2).

4. page 3, line 3: The sentence of “anti-diabetic medication lowers the risk of complications (ref 6)” is not appropriate, since the research compares 2 types of anti-diabetic medications.

#Thank you pointing this out. The reference was indeed incorrect. We removed this reference and instead refer to: Intensive blood-glucose control with sulphonylureas or insulin compared with conventional treatment and risk of complications in patients with type 2 diabetes (UKPDS 33). UK Prospective Diabetes Study (UKPDS) Group. Lancet. 1998 Sep 12;352(9131):837-53.

5. page 4, line 18: Does this mean that patients with the use of insulin within 180 days (2.5%) were not regarded as type 2 diabetes? This classification is clearly not appropriate.

#Your interpretation of the sentence is correct. Various methods exist to exclude insulin-dependent diabetes. One method (which was used) in this study, is to exclude those patients who start insulin within a certain period after diagnosis (here: 180 days) and who continue insulin treatment during a distinct period after diagnosis (here: the observation period). In our study eighteen of the 649 patients (2.5%) in the intervention group started insulin within 180 days after diagnosis. Insulin was discontinued for two of these patients during the observation period. Thus, at least 633 (97.5%) patients were considered to have type 2 diabetes. Olivarius NF, Beck-Nielsen H, Andreasen AH, Hørder M, Pedersen PA. Randomised controlled trial of structured personal care of type 2 diabetes mellitus. BMJ.2001 Oct 27;323(7319):970-5. 

6. page 5, line 2: Is the sentence “honeymoon phase” properly used?

#In our manuscript, we intended to describe the early improvement of glycemic control in our study population with newly diagnosed type 2 Diabetes, irrespective of the treatment modality. The term honeymoon phase is usually reserved for type 1 diabetes, indicating a transient restoration of the beta cell function following the initiation of insulin therapy in type 1 diabetes. Whether an analogous phenomenon exists in type II DM is debated (see reference 14: Retnakaran R. Novel Strategies for Inducing Glycemic Remission during the Honeymoon Phase of Type 2 Diabetes. CanJDiabetes. 2015;39 Suppl 5:S142-S7. doi: S1499-2671(15)00563-8). If the term is appropriate here is open to discussion: we therefore decided to remove the term (and reference 14) and rephrased the sentence as follows:

* “In general, participants experienced an improved glycemic control with a more or less pronounced remission of hyperglycaemia during the first year after diabetes diagnosis.”( l.79-80)

7. page 5, definition of outcomes line 1: Please clarify Diabetes-related endpoint and Diabetes-related mortality in this Section, although they are listed in Supplemental Table 2.

#We briefly explain the definition of the diabetes-related end-points and diabetes-related mortality in the revised version of the text and refer to table S1 for a more extended definition. 

* “The outcomes were all-cause mortality, diabetes-related mortality (i.e. cardiovascular mortality, death due hypo- and hyperglycemia, and sudden death) myocardial infarction, stroke, peripheral vascular disease, and microvascular disease, any diabetes-related endpoint (i.e. presence of any of the aforementioned categories) (please see table S1 for the exact definitions of the different outcomes).” (l. 89 - 93)

8. page 7, line 5: What does it mean by “HbA1c was, on average, normalized”? Please clarify.

#We meant to convey that the average HbA1c value was below the critical threshold of 7%. We decided to rephrase this sentence as follows: 

* “One year after diagnosis, HbA1c levels were on average lower than at time of diagnosis.” (l.142-143)

9. page 8, line 5, Supplemental table 2: The authors only stated the impact of insulin therapy on diabetes-related death. The effect of oral therapy on myocardial infarction is also significant, with significant interaction. This should also be described and discussed.

#We included the following sentence in the results section of the manuscript: 

* “Moreover, oral medication was a risk indicator for later myocardial infarction when compared to diet only and insulin therapy.”(l.177-179)

10. page 8, last line: The reviewer is against the idea that “the reduction in macrovascular disease was not attributable to poor HbA1c control,” since this may be the result of “legacy effect”. Please clarify.

#We agree with your suggestion. We included the legacy effect as a possible explanation for the long-term reduction of macrovascular morbidity. We rewrote the section as follows:

* “Thus, the present observational study suggest that the reduction in macrovascular disease may not attributable to quality of HbA1c control during the intervention period, but may be due to other elements of the intervention like blood pressure control. Alternatively, the lower macro-vascular morbidity in the DCGP study at 19 years may the results of the legacy effect, the long-term health benefits of early glycemic control.” (l.194-199)

11. page 9, line 14, ref 19: This reference dealt with microvascular complications besides amputation. Both of these findings should be discussed. (p.10; l. 211-212)

#Thank you pointing that out. We have included all the end-points from the meta-analysis in the current version of the manuscript: micro-, macrovascular morbidity and amputations. “A meta-analysis demonstrated that intensive treatment of Type 2 DM lowers the risk of microvascular complications as well as the risk of amputation of lower extremities. However, this meta-analysis failed to show an effect on macro-vascular morbidity.”

---

## [Decision Letter · Decision Letter 1]

31 Oct 2019

Changes in HbA1c during the first six years after the diagnosis of Type 2 diabetes mellitus predict long-term microvascular outcomes

PONE-D-19-16655R1

Dear Dr. Rozing,

We are pleased to inform you that your manuscript has been judged scientifically suitable for publication and will be formally accepted for publication once it complies with all outstanding technical requirements.

With kind regards,

Tatsuo Shimosawa, M.D., Ph.D.

Academic Editor

PLOS ONE

Additional Editor Comments (optional):

Reviewers' comments:

Reviewer's Responses to Questions

**Comments to the Author**

1. If the authors have adequately addressed your comments raised in a previous round of review and you feel that this manuscript is now acceptable for publication, you may indicate that here to bypass the “Comments to the Author” section, enter your conflict of interest statement in the “Confidential to Editor” section, and submit your "Accept" recommendation.

Reviewer #3: All comments have been addressed

2. Is the manuscript technically sound, and do the data support the conclusions?

Reviewer #3: Yes

3. Has the statistical analysis been performed appropriately and rigorously? 

Reviewer #3: Yes

4. Have the authors made all data underlying the findings in their manuscript fully available?

Reviewer #3: Yes

5. Is the manuscript presented in an intelligible fashion and written in standard English?

Reviewer #3: Yes

6. Review Comments to the Author

Reviewer #3: (No Response)

7. PLOS authors have the option to publish the peer review history of their article (what does this mean?). If published, this will include your full peer review and any attached files.

Reviewer #3: No

---

## [Editor Report · Acceptance letter]

19 Nov 2019

PONE-D-19-16655R1 

Changes in HbA1c during the first six years after the diagnosis of Type 2 diabetes mellitus predict long-term microvascular outcomes 

Dear Dr. Rozing:

I am pleased to inform you that your manuscript has been deemed suitable for publication in PLOS ONE. Congratulations! Your manuscript is now with our production department. 

With kind regards,

on behalf of

Prof. Tatsuo Shimosawa 

Academic Editor

PLOS ONE